# Rapid Start-Up Characteristics of Anammox under Different Inoculation Conditions

**DOI:** 10.3390/ijerph20042979

**Published:** 2023-02-08

**Authors:** Qiong Tan, Suhui Xia, Wenlai Xu, Yue Jian

**Affiliations:** 1Chongqing Academy of Animal Sciences, Chongqing 402460, China; 2National Center of Technology Innovation for Pigs, Chongqing 402460, China; 3State Key Laboratory of Geohazard Prevention and Geoenvironment Protection, Chengdu University of Technology, Chengdu 610059, China; 4Key Laboratory of Environmental and Applied Microbiology, Chinese Academy of Sciences, Chengdu 610041, China

**Keywords:** anammox, anaerobic granular sludge, starvation shock, start-up

## Abstract

The long multiplication time and extremely demanding enrichment environment requirements of Anammox bacteria (AAOB) have led to difficult reactor start-ups and hindered its practical dissemination. Few feasibility studies have been reported on the recovery of AAOB activity initiation after inlet substrate disconnection caused by an unfavorable condition, and few factors, such as indicators of the recovery process, have been explored. Therefore, in this experiment, two modified expanded granular sludge bed reactors (EGSB) were inoculated with 1.5 L anaerobic granular sludge (AGS) + 1 L Anammox sludge (AMS) (R1) and 2.5 L anaerobic granular sludge (AGS) (R2), respectively. After a long-term (140 days) starvation shock at a high temperature (38 °C), the bacteria population activity recovery experiments were conducted. After 160 days, both reactors were successfully started up, and the total nitrogen removal rates exceeded 87%. Due to the experimental period, the total nitrogen removal rate of R2 was slightly higher than that of R1 in the final stage. However, it is undeniable that R2 had a relatively long activity delay during startup, while R1 had no significant activity delay during startup. The sludge obtained from R1 had a higher specific anammox activity (SAA). Analysis of the extracellular polymer substances (EPS) results showed that the extracellular polymer content in R1 was higher than that in R2 throughout the recovery process, indicating that R1 had higher sludge stability and denitrification performance. Scanning electron microscopy (SEM) analysis showed that more extracellular filamentous bacteria could be seen in the R1 reactor with better morphology of Anammox bacteria. In contrast, the R2 reactor had fewer extracellular hyphae and micropores as a percentage and higher filamentous bacteria content. The results of microbial 16SrDNA analysis showed that R1 used AAOB as inoculum to initiate Anammox, and the reactor was enriched with Anammox bacteria earlier and in much greater abundance than R2. The experimental results indicated that inoculating mixed anaerobic granular sludge and Anammox sludge to initiate an anammox reactor was more effective.

## 1. Introduction

Anaerobic ammonia oxidation (Anammox) is a new technology developed recently for treating high ammonia nitrogen wastewater. The Anammox reaction is a process in which anaerobic ammonia-oxidizing microorganisms oxidize NH_4_^+^-N or NH_3_ to N_2_ using NO_2_^−^-N as an electron acceptor in a low or anaerobic environment (Equation (1)) [1]. The reaction does not require the provision of oxygen or adding an organic carbon source, which can significantly reduce energy consumption [2]. At the same time, no greenhouse gases such as CO_2_ and N_2_O are produced, making it an environmentally friendly wastewater treatment technology. The process has obvious advantages over conventional nitrification-denitrification processes.
NH_4_^+^ + 1.32NO_2_^−^ + 0.066HCO_3_^−^ + 0.13H^+^→0.066CH_2_O_0.5_N_0.15_ + 1.02N_2_ + 0.26NO_3_^−^ + 2.03H_2_O(1)

However, the long multiplication time of Anammox bacteria (AAOB) and the extremely demanding enrichment environment has led to difficult reactor start-up and hindered its practical dissemination. The right type of reactor not only helps to improve the start-up operational performance of the Anammox process but also helps to expand the engineering application of the process. Many reactors have been used to successfully start up Anammox processes, commonly including sequencing batch reactors [1], granular sludge reactors [3], biofilm reactors [4], membrane bioreactors [5], and other complex reactors [6]. Notably, the type of inoculated sludge is one of the critical factors for Anammox reactor start-up. Familiar sources of inoculated sludge include activated sludge, anaerobic granular sludge, and denitrifying sludge [7]. Studies have shown that direct inoculation of Anammox seed sludge is the most efficient method for rapid process start-up. However, AAOB is generally expensive and impractical to purchase in large quantities, so there is an urgent need to develop the most efficient sludge preservation method for the Anammox process during industrial production, transportation, and storage. Previous research on optimally storing anaerobic granular sludge experiments has focused on conditions such as low temperatures, short periods, and the addition of substrates or protectants. Few studies have been reported on the feasibility of the recovery of the bacterial population and comparative studies of different sludge start-ups after intermittent shutdowns, equipment maintenance, natural disasters, and other unexpected conditions that bring irresistible external resistance leading to prolonged reactor outages, causing long-term starvation of AAOB and affecting their activity [8,9] in engineering practice.

Given the above situation, this paper proposes to select anaerobic granular sludge, which is widely available in nature, as inoculum sludge, and inoculate 1.5 L anaerobic granular sludge (AGS) + 1 L Anammox sludge (AMS) (R1), and 2.5 L anaerobic granular sludge (R2) in two modified EGSB reactors to investigate the start-up recovery performance after long-term starvation shock. The purposes of this study were to (1) explore the feasibility of start-up recovery after an adverse shock and (2) compare the differences between the two reactors (inoculated with a single AGS and inoculated with a small amount of AMS). The results of this study are expected to facilitate the application of Anammox process for the treatment of industrial wastewater under long-term starvation conditions.

## 2. Materials and Methods

### 2.1. Seeding Sludge

Typical anaerobic granular sludge was used as the initial inoculum sludge purchased from China Water and Li Environmental Technology Co. This sludge had a charcoal black color, a volatile suspended solids (VSS) content of 68.7 g/L, a volatile substance content (VSS/SS) of 76%, a water content of 91%, and an adequate particle size of 83%. Anammox bacteria were used as the inoculum strain, and the AAOB were collected from the reaction column in the laboratory. The bacterium was entirely shaped and red. The inoculation volume ratio of AGS to AMS was 3:2 for the R1 reactor, and only anaerobic granular sludge was inoculated for R2. In both cases, the inoculum volume of both reactors was 2.5 L. After inoculation, the mixed liquor suspended solids (MLSS), and the mixed liquor volatile suspended solids (MLVSS) in both reactors were approximately 4.0 g/L and 3.04 g/L, respectively.

### 2.2. Feeding Water

In this study, synthetic wastewater was used, composed of nitrogen and inorganic carbon sources [2]. NH_4_^+^-N and NO_2_^−^-N were provided by two compounds, NH_4_Cl and NaNO_2_, respectively. The trace elements included EDTA (15 g L^−1^), FeSO_4_·7H_2_O (5 g L^−1^), H_3_BO_4_ (0.014 g L^−1^), MnCl_2_·4H_2_O (0.99 g L^−1^), CuSO_4_·5H_2_O (0.025 g L^−1^), ZnSO_4_·7H_2_O (0.43 g L^−1^), NiCl·6H_2_O (0.19 g L^−1^), NaMoO_4_·2H_2_O (0.22 g L^−1^), NaSeO_4_·10H_2_O (0.123 g L^−1^), CoCl_2_·6H_2_O (0.3 g L^−1^), and NaWO_4_·2H_2_O (0.664 g L^−1^). The synthetic wastewater was purged with N_2_ for about 20 min to obtain anaerobic conditions, and its dissolved oxygen concentration was controlled within 0.2 mg/L.

### 2.3. Experimental Setup

Two modified expanded granular sludge beds (EGSB) were used as reactors for Anammox with the structure shown in Figure 1. The reaction zone of each reactor is 750 mm high, with an inner diameter of 102 mm and an adequate volume of 4.5 L. A thermostatic water heating device is installed outside each reactor to stabilize the reactor’s operating temperature at 35 ± 5 °C. The reaction zone of each reactor was wrapped with black tin foil to avoid the effect of light on AAOB.

### 2.4. Analysis Methods

Inlet and outlet water samples were taken daily for NH_4_^+^-N, NO_2_^−^-N, and NO_3_^−^-N content determination. Before chemical analysis, water samples were filtered with a 0.45 μm microfilter, and water quality and MLVSS were determined according to standard methods [10]. DO was determined using a portable dissolved oxygen meter (607 A portable DO meter); pH was determined using a portable pH meter (SX-620 portable pH meter). Extracellular polymers (EPS) were extracted from sludge samples by a modified thermal extraction method [11]. The protein (PN) and polysaccharide (PS) contents were determined by the total protein quantification kit (BCA method) and phenol-sulfuric acid colorimetric method [12]. Specific anammox activity (SAA) was measured and calculated according to the method described in the literature [13]. The morphology of the sludge samples was observed using a digital camera (Nikon D3500, Nikon Corporation: Tokyo, Japan) and a microscope (LEICA DMLB, Leica Camera AG: Wetzlar, Germany). Their surface structure was explored by scanning electron microscopy (SEM) (Primsa E, Thermo Fisher Scientific: Waltham, MA, USA). The analytical methods and the instruments or reagents used are shown in Table 1.

### 2.5. High-Throughput Sequencing

The microbial communities of the sludge samples were analyzed by high-throughput 16S rRNA amplicon sequencing. During the run, 10 g of sludge samples (numbered A1, A2, A3, A4, B1, B2, B3, and B4) were taken from the bottom of the two reaction zones on days 1, 30, 120, and 160, respectively. Samples were submitted to Shanghai Meiji Biomedical Technology Co. (Shanghai, China). Then 515FmodF (5′-GTGYCAGCMG CCGCGGTAA-3), and 806modR (5′-GGACTACNVGGGTWTCTAAT-3′) were used for PCR amplification and library construction, and gene sequence analysis was performed on the Illumina Miseq platform. The Silva database (SSU128) was used to perform operational taxonomic unit (OUT) clustering of species with more than 70% sequence similarity, which has a 70% comparison threshold.

## 3. Results and Discussions

### 3.1. Nitrogen Removal Performance Comparison

The denitrification performance of the two reactors is shown in Figure 2.

In phase I (cell lysis phase; R1: 1–5 d; R2: 1–4 d), the concentration of NH_4_^+^-N in the effluent water of both R1 and R2 was higher than that of the influent water (Figure 2a). Due to the sudden change in environmental conditions, most inoculated microorganisms could not adapt to the anaerobic environment and inorganic conditions and autolyzed, releasing large amounts of COD and NH_4_^+^-N from bacteria [14], and thus the effluent NH_4_^+^-N increased dramatically. As anaerobic granular sludge is less flexible in adapting to its environment and is susceptible to higher shear sensitivity due to factors such as surface charge and EPS [15], anaerobic granular sludge is more prone to stripping, leading to the massive dissolution of organic matter from its anaerobic surface. Therefore, the NH_4_^+^-N content in R2 effluent is higher than in R1.

In phase II (lag phase; R1: 6–31 d; R2: 5–62 d), the removal rate of NH_4_^+^-N from R1 effluent gradually exceeded 90%, and the performance stabilized (Figure 2a). Compared to R1, the NH_4_^+^-N removal efficiency of the R2 reactor was unstable and showed a decreasing trend in NH_4_^+^-N removal efficiency on day 42. The shock resistance performance of R2 was lower than that of R1 due to the free ammonia (FA) released due to cell lysis in stage I which inhibited the Anammox activity to some extent [16,17]. The NO_2_^−^-N removal rate of the two reactors was good at the early stage of the startup, and some heterotrophic denitrifying bacteria were present in the inoculated mixed sludge, which were able to use the organic matter remaining in the sludge and produced by the endogenous metabolism of other microorganisms for denitrification with NO_2_^−^-N. Due to the continuous dilution of the influent water, the organic matter in the reactor was gradually decreasing, thus decreasing the ability to reduce denitrification activity. R1 and R2 showed a decrease in NO_2_^−^-N removal at this stage (Figure 2b). Compared with R2, the lag period of R1 was 31 days earlier, and the denitrification performance of R1 was more stable, indicating that inoculation of a small amount of AAOB could shorten the lag period and improve the performance stability of the system.

In phase III (transition phase; R1: 32–117 d; R2: 63–125 d), the influent nitrogen loading rate (NLR) increased from 0.06 kg/(m^3^·d) to 0.47 kg/(m^3^·d). In this phase, the feed water content of NH_4_^+^-N was gradually increased, both R1 and R2 maintained high NH_4_^+^-N removal efficiency (Figure 2a). The hydraulic residence time (HRT) of the R1 reactor was unchanged in the early stage and shortened to 16h at 110d, the return valve was opened, and the return ratio was set to 100%. The nitrogen removal performance of the R2 reactor was stable and showed apparent characteristics of an Anammox reaction, i.e., the simultaneous and equal proportional increase of NH_4_^+^-N and NO_2_^−^-N removal was accompanied by a gradual increase of NO_3_^−^-N production (Figure 2d) [18], and the maximum volumetric matrix nitrogen removal rate (NRR) was 0.47 kg N/(m^3^·d) (Figure 2e). This indicates that, to some extent, AAOB have gradually replaced some denitrifying bacteria and dominated the system. On day 121, the effluent ammonia-nitrogen and nitrite concentrations deteriorated rapidly due to the failure of the R2 reactor thermostat. On day 123, the thermostat returned to normal, and the NH_4_^+^-N and NO_2_^−^-N removal rates within the R2 reactor recovered rapidly, indicating that the reactor also operated stably during this period. At the end of stage III, total nitrogen removal efficiencies in R1 and R2 reached 84.2% and 81.6%, respectively (Figure 2c), which were close to the theoretical removal rates of total nitrogen for Anammox.

In phase IV (activity elevation phase; R1: 118–160 d; R2: 126–160 d), the influent NLR increased from 0.47 kg/(m^3^·d) to 1.45 kg/(m^3^·d) (Figure 2e). The R1 return valve was opened, and the return ratio was set to 100% on day 120; on day 140, the HRT was further shortened to 8 h, and the return ratio was set to 200%. The continuous increase of influent nitrogen load and hydraulic shear promoted the granulation process and rapid enrichment of AAOB, and the Anammox activity was further improved. By day 151, the TN removal rate was 87.44%, the ratio of NO_2_^−^-N removal to NH_4_^+^-N removal (Rs) was 1.31, and the ratio of NO_3_^−^-N production to NH_4_^+^-N removal (Rp) was 0.27, which was close to the theoretical value proposed by Stous et al. [1]. The influent volumetric load of the R2 reactor was gradually increasing, but the performance of the reactor was very stable at this stage. The effluent NH_4_^+^-N and NO_2_^−^-N concentrations were maintained in a low range, and the NO_3_^−^-N production increased with the Anammox reaction activity (Figure 2d). At the end of the whole reaction stage, the TN removal efficiencies of R1 and R2 reached 87.5% and 90.2%, respectively (Figure 2c), which were much higher than those of other researchers [19].

### 3.2. Stoichiometric Ratio

The stoichiometric ratio of substrate to product visualizes the Anammox reaction process. The theoretical stoichiometric ratios of Rs (∆NO_2_^−^-N/∆NH_4_^+^-N) and Rp (∆NO_3_^−^-N/∆NH_4_^+^-N) were 1.32 and 0.26, respectively [1]. Rs and Rp values often do not strictly conform to theoretical values due to differences in substrate loading, reactor configuration under start-up conditions, and functional bacteria abundance. Miao et al. chose anaerobic granular sludge to initiate Anammox and obtained a stoichiometric ratio of 1:1.26:0.44 [20]. The magnitude of the deviation of Rs and Rp from theoretical values can also be used to judge which reactions are performed within the system based on. In this experiment, the start-up process of the R1 system was complex, with negative values of Rs appearing at the beginning of the start-up period due to the higher effluent ammonia nitrogen mass concentration than the influent, followed by a slightly higher Rs value than the theoretical value and an Rp value slightly less than 0.26 during the active hysteresis phase due to the dominance of denitrification (Figure 2f). With the dominance of Anammox, the stoichiometric ratio of the reactor was 1:(1.32 ± 0.02):(0.27 ± 0.01). The Rs value of the R2 system was also much higher than the theoretical value at the initial stage of a startup because of the dominance of denitrification. However, with the increase of AAOB abundance, the Anammox activity in the reactor increased significantly. The Anammox reaction gradually took dominant, and the Rs and Rp values were close to the theoretical values by the 90th day of a startup, after which the R2 system maintained efficient denitrification efficacy with a further increase in substrate loading. The stoichiometric ratio of the reactor during the stabilization phase was 1:(1.32 ± 0.02):(0.27 ± 0.01).

### 3.3. pH

The increase of effluent pH at the identical HRT and incubation conditions indicates that the performance of Anammox is enhancing [21]. Figure 3a shows the pH change curve of the influent and effluent water during the start-up of Anammox in the R1 reactor. At the initial stage of the startup, the pH was slightly higher than the inlet water, which was due to the mutual neutralization and offsetting of alkali production by denitrification and acid production by bacteriophage hydrolysis in the reactor [22], while after the end of the autolysis period of the bacteriophage, the pH increment gradually increased due to the manifestation of Anammox activity and denitrification, and then showed a slight decrease with the decrease of denitrification activity; with the increase of AAOB activity, the Anammox reaction dominated, and the pH value of effluent was significantly more extensive than that of influent, and its increment showed a rising trend and gradually stabilized. The monitoring data of the last stage (HRT = 8 h) of the R1 start-up process were selected. The product of inlet and outlet ∆pH difference and influent water volume was used as the horizontal coordinate. The total nitrogen removal (NRR) was used as the vertical coordinate to make a graph (Figure 3b). There was an excellent linear relationship between the two. The influent pH of the R2 reactor was controlled to be within 7.8 to 8.2, and the effluent pH was between 8.0 and 8.7. Throughout the start-up period, the effluent pH of the R2 system was more significant than the influent pH (Figure 3c). In contrast, during the active stabilization period, the difference between the reactor inlet and outlet pH gradually stabilized to about 0.5 as the Anammox reaction dominated. Similar to R1, there was also an excellent linear relationship between the product of ∆pH and the inlet water Q and the total nitrogen removal load (Figure 3d). This indicates that the operating performance of both reactors was relatively stable at the end of the startup, and the Anammox denitrification efficiency was good [23].

### 3.4. Microstructure Morphology and SEM Analysis

As seen in Figure 4a, the inoculated AGS had a clear boundary and dense structure, similar to the observations made by other researchers [24,25]. After 100 days of domestication, they were removed from the bottom of the reactor and observed using microscopy at 40× and 100× magnification, respectively. The observations showed that the proportion of granular sludge with yellowish-brown color increased in the R1 and R2 reactors, which proved that the color change of granular sludge was a surface-to-surface process, presumably since the surface layer of granular sludge was more accessible to the sufficient substrate for metabolism, which was the same as the results of Liu et al. [26].At the end of the domestication period, microscopic observation of the samples retaken revealed that the color of granular sludge in both R1 and R2 reactors was predominantly reddish-brown, containing very little black granular material, which was presumed to be inorganic salt precipitates, and the formation of inorganic precipitates in granular sludge would accelerate the process of sludge granulation [27].

The surface of the mature granular sludge was further observed by scanning electron microscopy (SEM), as shown in Figure 4b. The surface of the granular sludge is rough with a large number of “micropores”, which are pore channels for the migration and transformation of substrate and gas molecules and help to improve the mass transfer efficiency of the sludge [28]. The higher the EPS content, the easier it is to promote cell coalescence into clusters. Many filamentous bacteria were also observed attached to the surface of the granules, with a length of about 2 μm. R2 was similar to the R1 sludge composition, but compared to the R1 reactor, fewer spherical bacteria were observed inside the R2 reactor granular sludge. Some of the spherical bacteria were notched in a concave shape, with a more fragmented degree of aggregation and a relatively small amount of reticulation. However, the sludge structure was dense and compact, with less pore area, and most of the filaments and short rods were connected to the sludge. SEM analysis showed that inoculation of anaerobic granular sludge and anaerobic ammonia-oxidizing composite sludge had some effects on the sludge morphology and abundance of AAOB, and more extracellular mycelia could be seen in the R1 reactor. The abundance of AAOB was higher, and the morphology was better. In contrast, the R2 reactor had less extracellular filamentous bacteria and microporous percentage and higher filamentous bacteria content.

### 3.5. Specific Anammox Activity

As shown in Figure 5a, after a long period of high-temperature starvation storage, the Anammox granular sludge lost most of its activity, and the specific Anammox activity (SAA) values in both reactors were shallow. After a period of incubation, the specific activity of Anammox microorganisms was significantly improved. After 100 and 160 days of incubation, the specific anammox activity of Anammox granular sludge in R1 increased from 0.0048 g g^−1^d^−1^ at the beginning to 0.278 g g^−1^d^−1^ and 0.466 g g^−1^d^−1^, respectively; the specific anammox activity of sludge in R2 increased from 0.0052 g g^−1^d^−1^ at the inoculation period after 100 and 160 days of incubation. Further, 0.0052 g g^−1^d^−1^ in the inoculation period to 0.24 g g^−1^d^−1^ and 0.397 g g^−1^d^−1^, respectively, with much higher variation than other researchers [29]. After a period of enrichment culture, the specific anammox activity of the strains in the R1 reactor was 13.67% and 14.81% higher than that in the R2 reactor, respectively, which may be due to the more complex microbial community composition of the anaerobic granular sludge plus AAOB inoculated in R1, resulting in more strains presenting SAA. Anammox bioreactor, after a long starvation shock, restoring its performance is a relatively easy task due to the long period of the AAOB era. Accelerating the rate of activity recovery and improving the competitiveness of AAOB during the start-up recovery phase is relatively difficult.

### 3.6. Extracellular Polymers

Extracellular polymers (EPS) is mainly composed of Protein (PN) and Polysaccharide (PS), which contributes significantly to improving the stable expression of bacterial genes and enhancing the adaptability of AAOB to the external environment [21]. The higher the EPS content of granular sludge, the easier it is for anaerobic ammonia-oxidizing microorganisms to coalesce and form polymers. The higher the EPS content, the better the stability of the sludge [30] and the more stable the denitrification performance. Throughout the recovery process, the EPS content in R1 was higher than in R2 (Figure 5b), indicating that R1 had higher sludge stability and denitrification performance. During the observation period, PN was the main factor contributing to the increase of EPS, and PS was relatively stable during sludge granulation. On days 40–60, EPS, as well as PN/PS, increased significantly in R1 and R2, presumably due to the massive production of some heterotrophic bacteria during this period due to lysis and death of poor substrate, as well as the synthesis of EPS by some microorganisms not adapted to the environment through endogenous metabolic stimulation to protect their cell structure [31]. On 100–160 d, the granular sludge enters the maturation phase, and the dense structure of mature sludge requires more EPS to maintain, resulting in less EPS secretion. The trend is similar to other researchers [32].PN/PS affects the settling performance of sludge [33]. The turning trend of PN/PS in R1 and R2 and the average particle size of sludge particles have a turning trend of PN/PS in R1 and R2 correlates with the average particle size of the sludge, and several turning points in the curve coincide with the turning period when more than half of the sludge in the reactor changes from small to large particles (Figure 5c). In general, the PN/PS ratio directly reflects the changes in hydrophobicity and electronegativity of the sludge: the cell surface charge is negatively correlated with the PN/PS ratio in EPS, the higher the PN/PS, the more electronegative the cells are, and the stronger their hydrophobicity and affinity [34]. Hydrophobic groups in EPS bridge with positively charged inorganic cations, facilitating the coalescence between sludge particles and maintaining their microscopic properties. Coalescence maintains their microstructure. Before the turnaround period, the particle size of granular sludge in the reactor was small, and there was a significant change in the particle size distribution. The sludge tends to secrete more EPS extracellularly while the PN/PS also increases, and the adsorption bridging effect is enhanced, which promotes the mutual aggregation between small particles. Furthermore, when the sludge particle size grows to a certain level, PN/PS gradually decreases with the decrease of EPS content, and the affinity between particles decreases, avoiding the formation of too large particles and making the settling performance decrease, thus affecting the reactor operation performance. In addition, it has been demonstrated that maintaining the particle structure of sludge is the key to achieving high SAA [35], and the high EPS and PN/PS values of R1 further explain the intrinsic reason for its higher SAA values relative to R2.

### 3.7. Microbial Community Analysis

The relative abundance of microbial communities at the genus level is summarized in Figure 6. *Candidatus Brocadia*, *Candidatus Jettenia* were the dominant AAOB. After stable operation of the reactor, *Candidatus Brocadia* was gradually eluted from the R1 reactor. *Candidatus Jettenia* was the only remaining dominant anaerobic genus in the reactor with 21.8%; meanwhile, *Candidatus Jettenia* was eventually detected at 14.04% in R2 after a stable operation. *Candidatus Brocadia*, which usually grows in adversity, was only enriched in R1 after a stable reactor operation [18]. *Candidatus Jettenia* in R1 *Candidatus Jettenia* was heavily enriched at the end of stage III of the reaction, much earlier than in the R2 reactor (where significant enrichment was detected at stage IV). *Denitratisoma* and *Thiobacillus* are common denitrifying bacteria [36]. With further AAOB activity, *Thiobacillus* was sieved out simultaneously in both reactors, but *Denitratisoma* maintained a specific abundance (2.17–2.54%). The results suggest that *Denitratisoma* is more viable than *Thiobacillus* under conditions suitable for the growth of AAOB. Possible reasons for this are: the abundance of *Denitratisoma* increased due to sufficient reaction substrate due to the decrease of heterotrophic microorganisms such as *norank_f__Anaerolineaceae*; mature AAOB granular sludge tended to secrete more EPS, the proteins of which could be used as a carbon source for denitrifying bacteria. Furthermore, the NO_3_^−^-N produced by the Anammox reaction also provides a substrate for *Denitratisoma* bacteria [37]. The methanogenic genus is the most abundant in R1 and R2. Based on the difference in substrate utilization, methanogenic bacteria can be classified into hydrotropic and acetic acid types [38]. *Methanobacterium* belongs to the former group and mainly uses H_2_ or CO_2_ as substrates [39] for metabolism. *Methanosaeta* belongs to the latter and mainly uses acetic acid as a substrate [40] for reproduction. The changes in the content of two archaea, methane bacteria (30.78% and 36.81% in R1 and R2 reactors, respectively) and methane bristlecone bacteria (5.18% and 12.88% in R1 and R2 reactors, respectively), at this time, roughly inferred that the Anammox system tends to be more towards hydrogenotrophic methanogenic bacteria.

## 4. Conclusions

After 160 days, both reactors were started up successfully, and the TN removal rate exceeded 87%. The activity lag period was relatively long during the start-up process of R2, and there was no significant activity lag period during the start-up process of R1, and the sludge obtained from R1 had a higher SAA. Throughout the recovery process, the EPS content was higher in R1 than in R2, and R1 had higher sludge stability. The R1 reactor could see more extracellular filamentous bacteria and higher abundance and better morphology of AAOB, while the R2 reactor had less extracellular filamentous bacteria and micropores as a percentage and higher filamentous bacteria content. The results of microbial 16SrDNA analysis showed that R1 used AAOB as inoculum to initiate Anammox, and the reactor was enriched with AAOB earlier and in much greater abundance than R2. The experimental results indicated that inoculation of mixed AGS and AMS to initiate an Anammox reactor was a more effective strategy. The results of this study are expected to promote the application of the Anammox process for treating industrial wastewater under long-term starvation conditions.

## Figures and Tables

**Figure 1 ijerph-20-02979-f001:**
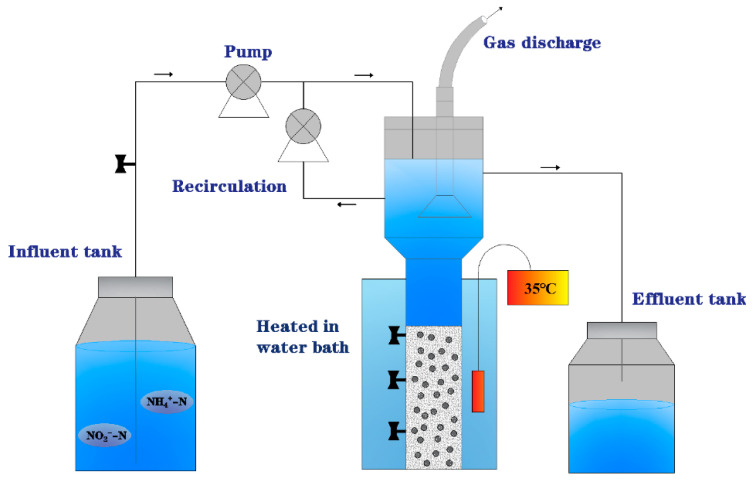
Schematic diagram of the expanded granular sludge beds (EGSB) used in the experiments.

**Figure 2 ijerph-20-02979-f002:**
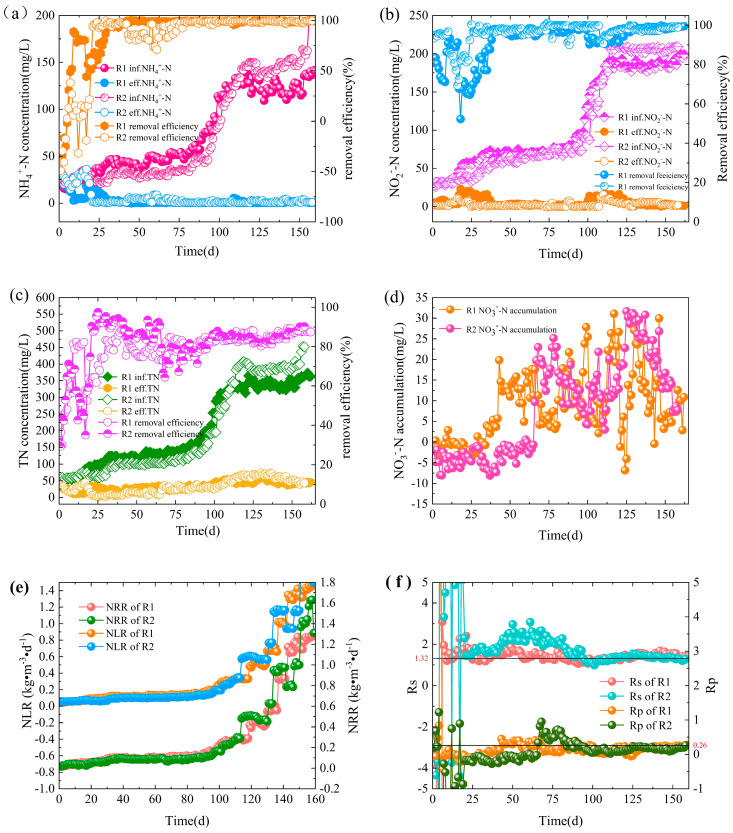
Long-term performance of nitrogen removal in the reactors. Concentrations and removal rates are shown for NH_4_^+^-N (**a**); NO_2_^−^-N (**b**); total nitrogen (TN) (**c**); and the ΔNO_3_^−^-N accumulations (**d**); variations of nitrogen loading rate (NLR) and nitrogen removal rate (NRR) (**e**) and of Rs (ΔNO_2_^−^-N/ΔNH_4_^+^-N) and Rp (ΔNO_3_^−^-N/ΔNH_4_^+^-N) (**f**).

**Figure 3 ijerph-20-02979-f003:**
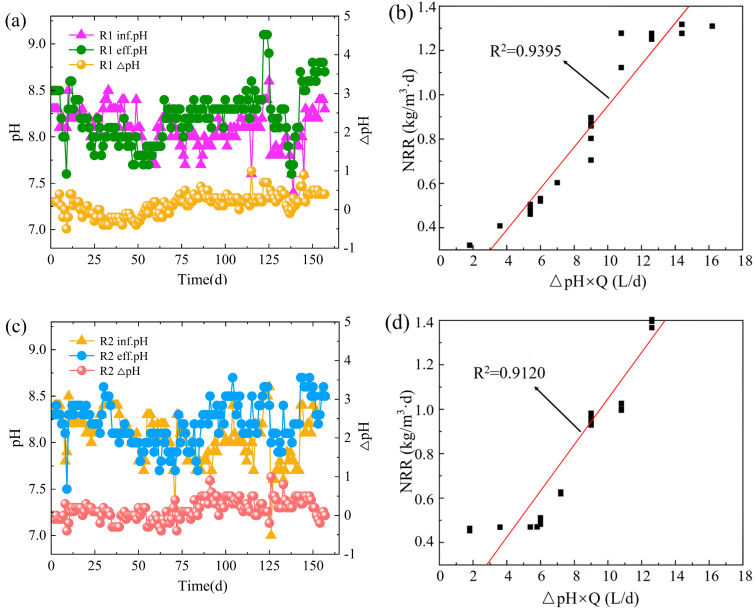
Changes in pH of R1 during start-up (**a**) and its relationship with nitrogen removal (NRR) (**b**); changes in pH of R2 during start-up (**c**) and its relationship with nitrogen removal (NRR) (**d**).

**Figure 4 ijerph-20-02979-f004:**
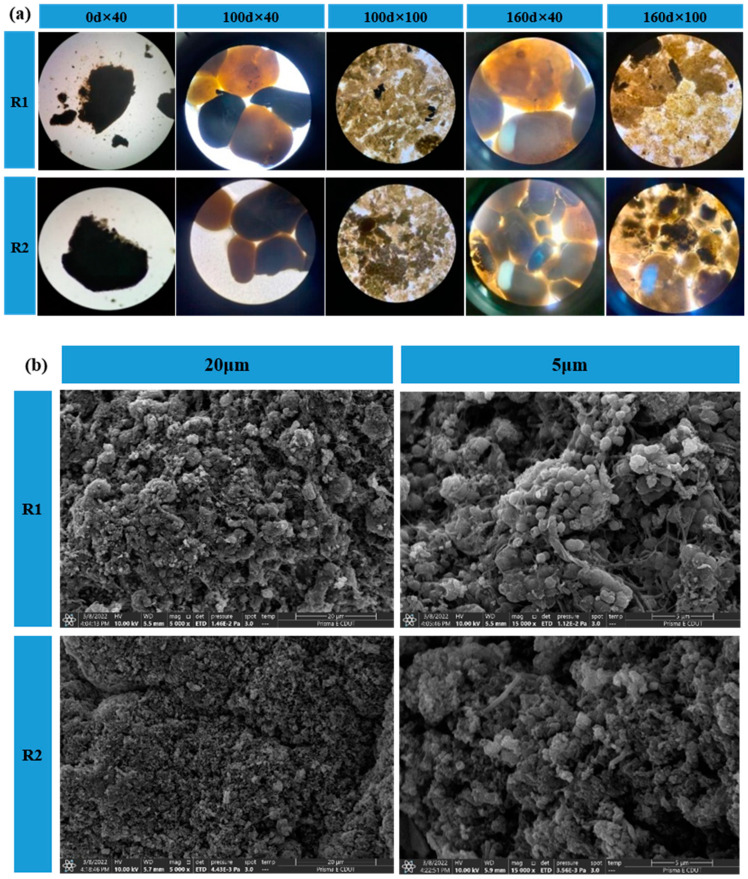
Microbial morphology analysis by (**a**) microscopic microscopy and (**b**) SEM. Pictures represent granular taken on day 0, day 100 and day 160, and floating granules taken on day 160.

**Figure 5 ijerph-20-02979-f005:**
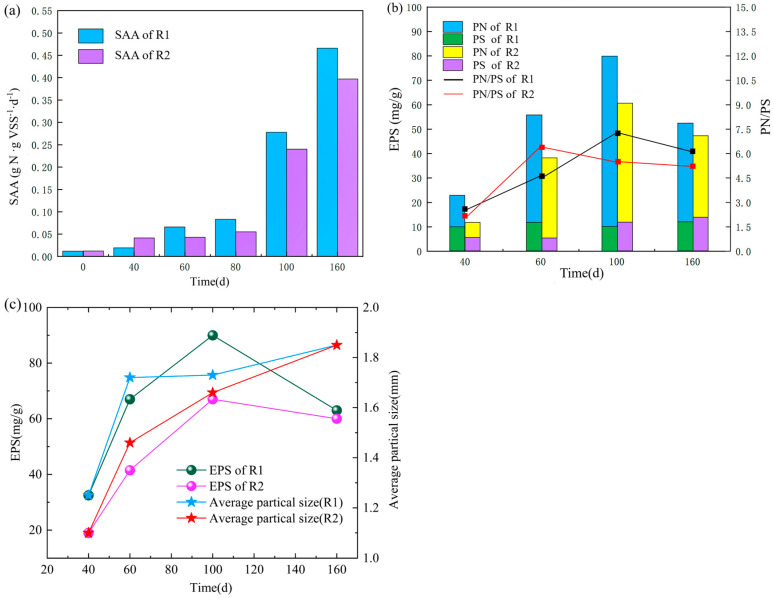
Changes of specific anammox activity (SAA) in R1 and R2 during the reaction phase (**a**); content of extracellular polymeric substances (EPS) and its fractions of protein (PN) and polysaccharide (PS) (**b**); relationship between average particle size and extracellular polymeric substances (EPS) during sludge granulation in R1 and R2 (**c**).

**Figure 6 ijerph-20-02979-f006:**
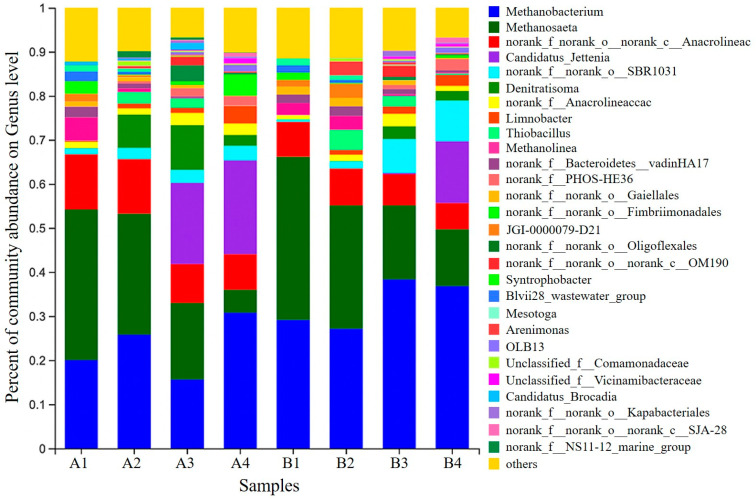
Microbial genus horizontal community structure distribution map; microbiological assays were performed on day 1, day 30, day 120, and day 160 for R1 and R2, with samples numbered A1, A2, A3, and A4 (R1); B1, B2, B3, and B4 (R2).

**Table 1 ijerph-20-02979-t001:** Analysis methods and equipment used for key indicators.

Analytical Projects	Analysis Methods	Instruments Used
NH_4_^+^-N	Nascent reagent spectrophotometry	V-1100D Visible Spectrophotometer, Shanghai Meipu (Shanghai, China)
NO_2_^−^-N	N-(1-naphthyl)-ethylenediamine photometric method	V-1100D Visible Spectrophotometer, Shanghai Meipu (Shanghai, China)
NO_3_^−^-N	UV spectrophotometry	TU-1901 UV Spectrophotometer, Beijing Pu-Analysis (Beijing, China)
EPS Extraction	The sample was washed 3 times with pure water, then transferred to a 50 mL centrifuge tube, added 20 mL of pure water, heated at 80 °C f or 10 min, and centrifuged at 8× *g* for 20 min. The supernatant was filtered through a 0.45 µm filter membrane, and the filtrate was the EPS extract, which was stored at −20 °C.	BIO-RAD Burroughs Enzyme Labeler (Hercules, CA, USA)
PN	BCA Protein Reagent Kits (C503051)	BIO-RAD Burroughs Enzyme Labeler (Hercules, CA, USA)
PS	phenol-sulfuric acid method	BIO-RAD Burroughs Enzyme Labeler (Hercules, CA, USA)
MLVSS	Burning weight reduction method	Muffle furnace (KSW-6-12) (Beijin, China)
Surface structure	The samples were first washed 3 times with deionized water, then fixed with glutaraldehyde at a concentration of 2.5% in a refrigerator at 4 °C for 4 h, and the resulting mud-water mixture was then rinsed 3 times in 1 mol/L phosphate buffer solution, then dehydrated with ethanol solutions at concentrations of 10%, 20%, 40%, 60%, 80%, and 100% (a total of 3 times), followed by a 1:1 mixture of ethanol, isoamyl acetate, and pure Isoamyl acetate solution for 1 time, and finally dried and sprayed with gold.	SEM (Prisma, E) (Waltham, MA, USA)

## Data Availability

Not applicable.

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
