# Peer review of "Rapid Start-Up Characteristics of Anammox under Different Inoculation Conditions"

_ijerph, 2023, doi:10.3390/ijerph20042979_

Round 1
Reviewer 1 Report
The manuscript submitted for review concerns the issue of rapid start-up characteristics of anammox under different inoculation conditions. This is an important issue that requires research. The authors wrote the manuscript efficiently. However, I have some minor remarks: 1) statistical analyzes of the results are missing 2) graphs are not legible
Author Response
On behalf of all contributing authors, we would like to express our sincere appreciation to the reviewers for their constructive comments on our article entitled ‘Rapid start-up characteristics of anammox under different inoculation conditions’. These comments were valuable and helped to improve our article. We checked and corrected the grammar and spelling in the manuscript. Based on the comments of the reviewers, we have extensively revised the manuscript and added additional data to make our results convincing. Based on the suggestions, we have made extensive corrections to our previous manuscript, which are as follows.
1) statistical analyzes of the results are missing
This proposal is very meaningful to us. Our topic was to study the start-up performance of the anammox reaction in two reactors under different inoculation conditions. In the manuscript, a part of the statistical analysis was performed for some indicators, such as the pH variation during the start-up of the two reactors, the variation of EPS content, and the relationship between the variation of PN/PS and the average particle size of the sludge particles. The statistical analysis allowed us to find correlations between the indicators; for example, by the statistical analysis between the change in PN/PS and the average particle size of the sludge particles, we found several turning points in the PN/PS change curve that coincide with the turning period when more than half of the sludge in the reactor changes from small particles to large particles. For other indicators that were not subjected to extensive statistical analysis, we focused on the comparison of the two reactors, such as the denitrification performance, anaerobic ammonia oxidation activity, and microstructure morphology of the two reactors. Through the comparison we could find the trends and differences between the two reactors in these indicators, and through these we could see the start-up characteristics of the two reactors, which led to the conclusion that the R1 reactor has better start-up characteristics. According to this suggestion, we added statistical analysis of microorganisms in both reactors in section 3.7 of the manuscript. We found that the inoculation method of R1 was favorable to the rapid enrichment of AAOB flora and shortened the activity lag period. The abundance of AAOB flora in R1 was much larger than that in R2. This could make our results more convincing to the readers.
2) graphs are not legible
We apologize for not checking the quality of the charts due to our negligence. We have reworked and reuploaded the unclear diagrams in the manuscript, such as Figures 2, 3 and 5. We further optimized the legend and the description of the diagrams to make them more understandable to the readers.
Reviewer 2 Report
The submitted article (entitled: Rapid start-up characteristics of anammox under different inoculation conditions) is well-organised, informative and has potential applications. However, the following points should be addressed:
Minor issues
1- There are a few grammar and spelling mistakes.
2- The abstract must be started with the research question (the idea of the work).
3- The title of section 2.4 must be corrected to Analysis methods (Not methos)
Major issues
1- The novelty should be clearly stated in the abstract.
2- In the Materials and Methods
- Mention the country of the Water and Leaf Environmental Technology Co.
- Did you measure the characteristics of the AGS sample, or was it provided by the company?
- In section 2.4, please add the references of the used procedures/methods. For example, who recommended the use of a 0.45μm microfilter?
- In section 2.4, mention the specifications and models of the used camera and microscope.
4- More than 25% of the used references are older than 10 years. Replace the old ones where possible.
5- Some new studies might be of interest, such as:
* https://doi.org/10.1016/j.jwpe.2017.11.004
Author Response
On behalf of all contributing authors, we would like to express our sincere appreciation to the reviewers for their constructive comments on our article entitled ‘Rapid start-up characteristics of anammox under different inoculation conditions’. These comments were valuable and helped to improve our article. We checked and corrected the grammar and spelling in the manuscript. Based on the comments of the reviewers, we have extensively revised the manuscript and added additional data to make our results convincing. Based on the suggestions, we have made extensive corrections to our previous manuscript, which are as follows.
1- There are a few grammar and spelling mistakes.
We have tried our best to revise and embellish the article, these changes will not change the content and framework of the paper, thank you for your valuable comments.
2- The abstract must be started with the research question (the idea of the work).
This comment makes much sense to us. We have modified the abstract section by adding our ideas for this work at the beginning of the abstract. For example,’ The long multiplication time and extremely demanding enrichment environment requirements of anammox bacteria (AAOB) have led to difficult reactor start-up and hindered its practical dissemination.’
3- The title of section 2.4 must be corrected to Analysis methods (Not methos)
We are very sorry that this was due to our carelessness and we have corrected it accordingly.
Major issues
1- The novelty should be clearly stated in the abstract.
Based on the comments, we have added to the abstract content to enhance it in terms of illustrating the article's novelty, for example, ‘Few feasibility studies have been reported on the recovery of AAOB activity initiation after inlet substrate disconnection caused by an unfavorable condition, and few factors, such as indicators of the recovery process, have been explored.’
2- In the Materials and Methods
- Mention the country of the Water and Leaf Environmental Technology Co.
- Did you measure the characteristics of the AGS sample, or was it provided by the company?
- In section 2.4, please add the references of the used procedures/methods. For example, who recommended the use of a 0.45μm microfilter?
- In section 2.4, mention the specifications and models of the used camera and microscope.
Based on the valuable comments, we have made extensive changes to Section 2. The characteristics of the AGS samples were provided by the company from which we purchased them, and the sludge characteristics after inoculation were measured by us. In section 2.4, references to the experimental method have been added. The experimental method and the reagents used have been extensively supplemented. The specifications and models of the camera and microscope have also been standardized.
4- More than 25% of the used references are older than 10 years. Replace the old ones where possible.
This comment makes a lot of sense. We have corrected the number and quality of the references. The percentage of literature older than 10 years has decreased significantly to about 15%.
5- Some new studies might be of interest, such as:
* https://doi.org/10.1016/j.jwpe.2017.11.004
We appreciate the reviewer's suggestion, which can bring us new ideas. The SBR method is widely used to retain anaerobic ammonia oxidizing bacteria (AAOB) in anammox reactors, in which AAOB can be effectively retained in the form of anammox granular sludge. However, during the treatment of high ammonia nitrogen wastewater with anammox, AAOB generates a large amount of N2, which forms air pockets inside the granular sludge or adheres to the granular sludge surface, resulting in a decrease in granular sludge density, leading to sludge uplift and subsequent loss with the effluent. The innovative method in the literature has significantly improved the settling performance of SBR sludge while ensuring the treatment effect. This is a very good guideline for us. We can find new and more interesting ideas on this method and make it a reality, which can effectively promote the application of anammox action on SBR reactors.
Reviewer 3 Report
After reviewing the manuscript, which is generally devoted to the topical issue of nitrogen removal in anammox reactor, I can draw the following conclusions. Without any doubt it is an important topic and it has a high significance. In general, the manuscript is suitable for the aims and scopes of the journal IJERPH.
There are several comments on the text.
Lots of grammatical errors and typos throughout the text. Brackets with links must be separated from the text by spaces.
Authors should decipher abbreviations in the abstract. The term mycelium, which is suitable for fungi and actinomycetes, should be avoided.
The introduction should be supplemented with a large number of references to publications on similar problems of anammox bioreactors.
throughout the text it is worth checking and correcting indices in chemical formulas.
It is worth specifying for what type of waste it is planned to use the installation, origin, etc.
why did the authors not use instruments for nitrogen analysis? colorimetric methods do not always give accurate results.
In Figures 3 and 5 it is worth writing the legend more clearly, it is difficult to understand the meaning of a lot of abbreviations.
It is not very clear how the amount of EPS was determined (probably this is the mycelium you mentioned?)
Arguments about hydrophobicity should be confirmed by instrumental methods of analysis.
And probably the most importantly, the presence of anammox bacteria in activated sludge should be proved by at least some instrumental methods, FISH, real-time PCR or 16S genes analysis.
Author Response
On behalf of all contributing authors, we would like to express our sincere appreciation to the reviewers for their constructive comments on our article entitled ‘Rapid start-up characteristics of anammox under different inoculation conditions’. These comments were valuable and helped to improve our article. We checked and corrected the grammar and spelling in the manuscript. Based on the comments of the reviewers, we have extensively revised the manuscript and added additional data to make our results convincing. Based on the suggestions, we have made extensive corrections to our previous manuscript, which are as follows.
Lots of grammatical errors and typos throughout the text. Brackets with links must be separated from the text by spaces.
We have tried our best to improve the manuscript and made some changes to the manuscript. These changes will not influence the content and framework of the paper. The formatting in the article was also carefully checked and revised. I would very much like to receive your approval.
Authors should decipher abbreviations in the abstract. The term mycelium, which is suitable for fungi and actinomycetes, should be avoided.
Thanks to the reviewer for the professional suggestions, which were very enlightening. We have made changes based on this suggestion. All abbreviations in the abstract have been explained. At the same time, we have changed "mycelium" to "filamentous bacteria" in the manuscript.
The introduction should be supplemented with a large number of references to publications on similar problems of anammox bioreactors.
Based on the valuable comments, we have supplemented our introduction with the types of reactors associated with anaerobic ammonia oxidation reactions, for example, ‘Many reactors have been used to successfully start up anaerobic ammonia oxidation processes, commonly including sequencing batch reactors, granular sludge reactors, biofilm reactors, membrane bioreactors and other complex reactors.’ It has been augmented with a large body of relevant literature.
throughout the text it is worth checking and correcting indices in chemical formulas.
We apologize for the errors caused by our oversight, we have checked and corrected them.
It is worth specifying for what type of waste it is planned to use the installation, origin, etc.
For us, this is a very meaningful opinion. We designed our experiments with a relatively high influent nitrogen load, and the influent water used was laboratory-prepared water. The influent concentration exceeded most of the common water pollution indicators. Hence, our topic is more important to help accelerate anaerobic ammonia oxidation to treat high-concentration wastewater such as industrial wastewater. In the article, we have made additional explanations, for example, ‘The results of this study are expected to facilitate the application of anammox process for the treatment of industrial wastewater under long-term starvation conditions.’
why did the authors not use instruments for nitrogen analysis? colorimetric methods do not always give accurate results.
We used instruments for the detection of nitrogen during the experiment. Also, we have added the experimental methods, apparatus, and reagents used in the testing procedure in section 2.4.
In Figures 3 and 5 it is worth writing the legend more clearly, it is difficult to understand the meaning of a lot of abbreviations.
This advice is very instructive for us. We have reorganized the figures in the article, and the legend has been further enhanced to illustrate them.
It is not very clear how the amount of EPS was determined (probably this is the mycelium you mentioned?)
The amount of extracellular polymers (EPS) was derived from our measurements. The EPS content consisted of protein (PN) and polysaccharide (PS). EPS was first extracted from the sludge by thermal extraction, and the amount of protein and polysaccharide were measured separately, and the sum of the two was the amount of EPS. The experimental method is described in detail in section 2.4.
Arguments about hydrophobicity should be confirmed by instrumental methods of analysis.
This suggestion makes sense. However, we feel very sorry that the experiment has been stopped. We cannot measure the hydrophobicity and other cells' characteristics anymore. We found in our experiments that there is a correlation between the turning trend of PN/PS and the average particle size of sludge particles in both reactors, and several turning points in the curve coincide with the turning period when more than half of the sludge in the reactor goes from small to large particles. The hydrophobicity of sludge is related to the surface charge and the ionized groups on the sludge surface, which can increase the polar contact between EPS and water molecules. Therefore, the more the surface charge of the sludge, the lower the hydrophobicity, and the presence of hydrophobic substances such as proteins and fats in the EPS can make the sludge surface locally hydrophobic. Among them, amino acids with hydrophobic sites in the proteins play a dominant role in the hydrophobicity of the sludge surface. We cited the reference paper entitled "Effect of Carbohydrates to Protein Ratio in EPS on Sludge Settling Characteristics" by Hang-Sik Shin to illustrate the reasons for this phenomenon. This issue deserves to be explored further, as the settling performance of sludge is an important factor in reactor start-up, and we thank the reviewers for their valuable suggestions.
And probably the most importantly, the presence of anammox bacteria in activated sludge should be proved by at least some instrumental methods, FISH, real-time PCR or 16S genes analysis.
This comment is very important to us. We have added to the article in sections 2.5 and 3.7 about microbiological analysis measurements and analysis. From the microbiological analysis, we conclude that the inoculation of R1 favors the rapid enrichment of AAOB and shortens the activity lag period and that the abundance of AAOB colonies in R1 is much greater than in R2.
Reviewer 4 Report
Title: Rapid start-up characteristics of anammox under different inoculation conditions
The paper investigates the possibility of restarting the process of anaerobic digestion after a longer period of starvation, i.e. a break in the operation of the plant for the purpose of removing ammonia from wastewater. The research was carried out in two systems, one with anaerobic granular sludge with anammox sludge (R1) and the other only with anaerobic granular sludge (R2). Based on the examination of various process variables such as nitrogen removal, pH, SSA, EPS, morphological features, the authors conclude that the R1 system is more efficient.
In conclusion, this statement should be better justified since the TN removal efficiency is better with the R2 system.
I am also interested in whether the concentration of ammonia and other tested nitrogen compounds such as nitrates and nitrites has fallen below the limit values prescribed by law? Specified in paper.
The manuscript has many abbreviations that have never been explained.
It is not usual to use abbreviations in the Abstract.
The abbreviations 3.5 and 3.6 should not be used in the title.
Examples of abbreviations that are not explained: line 11, 62: EGSB (the explanation is only in line 84), line 14, 15, 17, 99, 135.
Chapter 2 needs to be improved. For example, in subsection 2.2. write units of measurement separately from the number. The R1 and R2 systems are not explained anywhere in this chapter, nor is the method of conducting the experiment, phases 1-6. Briefly describe the microscopic technique and the SEM technique used in sludge characterization.
The legends in Figures 2, 3, 5 are not legible, they need to be enlarged. It would also be good to enlarge the Figures and symbols.
In section 3.1. you list 6 phases - what about phase 4 and 5?
Author Response
On behalf of all contributing authors, we would like to express our sincere appreciation to the reviewers for their constructive comments on our article entitled ‘Rapid start-up characteristics of anammox under different inoculation conditions’. These comments were valuable and helped to improve our article. We checked and corrected the grammar and spelling in the manuscript. Based on the comments of the reviewers, we have extensively revised the manuscript and added additional data to make our results convincing. Based on the suggestions, we have made extensive corrections to our previous manuscript, which are as follows.
The paper investigates the possibility of restarting the process of anaerobic digestion after a longer period of starvation, i.e. a break in the operation of the plant for the purpose of removing ammonia from wastewater. The research was carried out in two systems, one with anaerobic granular sludge with anammox sludge (R1) and the other only with anaerobic granular sludge (R2). Based on the examination of various process variables such as nitrogen removal, pH, SSA, EPS, morphological features, the authors conclude that the R1 system is more efficient.
In conclusion, this statement should be better justified since the TN removal efficiency is better with the R2 system.
The reviewers' suggestions were very instructive. The results of the final total nitrogen appeared by chance because the experiment was forced to stop during the stabilization phase of the resumed experiment. We made changes to the abstract and conclusions, for example,‘Due to the experimental period, the total nitrogen removal rate of R2 was slightly higher than that of R1 in the final stage. However, it is undeniable that R2 had a relatively long activity delay during startup, while R1 had no significant activity delay during startup .’At the same time, we added the section on microbial analysis in chapter 3.7 of the manuscript. Through microbial community analysis, we found that the inoculation of R1 facilitated the rapid enrichment of AAOB flora and shortened the activity lag period, and the abundance of AAOB flora in R1 was much greater than that in R2, which can make the reader more convinced.
I am also interested in whether the concentration of ammonia and other tested nitrogen compounds such as nitrates and nitrites has fallen below the limit values prescribed by law? Specified in paper.
The effluent concentrations of NH4+-N and NO2--N have been reduced to below the legal limits, but the effluent concentrations of NO3--N are higher and exceed the legal limits. This is because, according to the reaction principle of anammox, a certain amount of NO3--N is produced during its occurrence, and the accumulation of NO3--N in the effluent is one of the important features of the anammox reaction. The removal of NO3--N is mainly through denitrification, which requires a large amount of carbon source. In our experiments, the feed water contains only the substrate required for the anammox reaction and does not contain a carbon source, so NO3--N accumulates continuously. Anammox is increasingly used but rarely used as the only removal process. It is often used with other processes to obtain better and more economical wastewater treatment results. For example, in the following paper, the authors used a combined PN/A + PD/A continuous flow process to treat mature waste leachate for nitrogen removal with good results. Our experiment aimed to investigate anammox's start-up characteristics under long-term starvation conditions, so NO3--N removal was not included as a condition of the experiment, and therefore the effluent of both NO3--N and TN was above the limits set by law.
https://doi.org/10.1016/j.biortech.2019.122483
The manuscript has many abbreviations that have never been explained.
We thank the reviewers for their careful examination. We are sorry that it was an oversight on our part that caused such a problem. Based on the comments, we checked all manuscript abbreviations and explained them when they first appeared.
It is not usual to use abbreviations in the Abstract.
We thank the reviewers for their professional advice. It will help us a lot in writing future articles. We have removed or explained the abbreviations that appear in the abstract.
The abbreviations 3.5 and 3.6 should not be used in the title.
We thank the reviewers for their professional comments. It can teach us a lot. Based on the comments, we changed the title of section 3.5 to "Specific anammox activity"; and section 3.6 to "Extracellular polymers".
Chapter 2 needs to be improved. For example, in subsection 2.2. write units of measurement separately from the number. The R1 and R2 systems are not explained anywhere in this chapter, nor is the method of conducting the experiment, phases 1-6. Briefly describe the microscopic technique and the SEM technique used in sludge characterization.
We are very sorry for the simple errors caused by our carelessness, and we have checked and corrected these errors. Also, additional explanations for the R1 and R2 systems are given in section 2.1 of the manuscript, for example, ‘The inoculation volume ratio of anaerobic granular sludge to anaerobic ammonia-oxidizing bacteria was 3:2 for the R1 reactor, and only anaerobic granular sludge was inoculated for R2. In both cases, the inoculum volume of both reactors was 2.5 L’.
We divided the whole process into four stages based on different characteristics of the anammox initiation process. In phase I, the concentration of NH4+-N in the effluent of both R1 and R2 was higher than that of the influent water. In phase II, the NH4+-N removal rate of both reactors stabilized. In phase III, the NLR of the incoming water was increased moderately. In phase IV, the NLR was increased significantly. We have summarized the generalized names of the four stages in the manuscript, cell lysis phase, lag phase, transition phase, and activity elevation phase. We have prefaced each stage with the characteristics of that stage.
In chapter 2.4 we have added the methods, instruments and reagents involved in the experimental procedure, and we have also added references to the experimental methods. We supplemented the microscope and SEM used in the sludge characterization with the model number and the specific experimental methods.
The legends in Figures 2, 3, 5 are not legible, they need to be enlarged. It would also be good to enlarge the Figures and symbols.
This observation is very meaningful and will be a great guide for our future papers. We have retouched and improved Figures 2 and 3 as well as Figure 5 by enlarging the legends and elaborating the meaning of the figures.
In section 3.1. you list 6 phases - what about phase 4 and 5?
In section 3.1, we divided the experiment into four phases according to the situation, and we mistakenly wrote "IV" as "VI", which we have corrected. We are very sorry that our carelessness led to such a simple mistake.
Round 2
Reviewer 3 Report
I have no significant comments on the revised version of the manuscript, the authors seriously corrected the text and made responses to all my comments. I believe that the manuscript in this form can be published in a journal.
Reviewer 4 Report
The authors improved the manuscript according to my suggestions and explained everything clearly in the answer, therefore my decision is acceptance of the manuscript in this form.